# Identification of immune-related biomarkers associated with allergic rhinitis and development of a sample diagnostic model

MaoMeng Wang[1,2,3,4], Shuang Wang[1,2,3,4], XinHua Lin[1,2,3,4], XiaoJing Lv[1,2,3,4], XueXia Liu[5*], Hua Zhang (ORCID)[1,2,3,4*]

**1** Qingdao University Affiliated Yantai Yuhuangding Hospital, Yantai, Shandong Province, China, **2** Department of Otorhinolaryngology, Head and Neck Surgery, Yantai Yuhuangding Hospital, Qingdao University, Yantai, Shandong, China, **3** Shandong Provincial Clinical Research Center for Otorhinolaryngologic Diseases, Yantai, Shandong, China, **4** Yantai Key Laboratory of Otorhinolaryngologic Diseases, Yantai, Shandong, China, **5** Shandong Stem Cell Engineering Technology Research Center, Affiliated Yantai Yuhuangding Hospital of Qingdao University, Yantai, China

* zhang0hua@163.com (HZ); xue6er6@163.com (XXL)

## Abstract

This study was designed to identify immune-related biomarkers associated with allergic rhinitis (AR) and construct a robust a diagnostic model. Two datasets (GSE5010 and GSE50223) were downloaded from the NCBI GEO database, containing 38 and 84 blood CD4 + T cell samples, respectively. To eliminate batch effects, the surrogate variable analysis (sva) R package (version 3.38.0) was employed, enabling the integration of data for subsequent analysis. Immune cell infiltration profiles were assessed using the Gene Set Variation Analysis (GSVA) R package (version 1.36.3). A gene co-expression network was constructed via the Weighted Gene Co-Expression Network Analysis (WGCNA) algorithm to identify disease-related modules. Differentially expressed genes (DEGs) were identified using the linear models for microarray data (limma) R package (version 3.34.7), followed by functional enrichment analysis using DAVID. Protein-protein interaction (PPI) networks were constructed based on the STRING database to highlight key genes. A diagnostic model was subsequently developed utilizing the Least Absolute Shrinkage and Selection Operator (LASSO) regression algorithm and Support Vector Machine (SVM) method, with its discriminative capacity assessed via Receiver Operating Characteristic (ROC) curves. A total of twenty-eight immune cell types were analyzed, revealing significant differences in eight types between the AR and control groups. Through WGCNA, three disease-related modules comprising 4278 candidate genes were identified. Differential expression analysis identified 326 significant DEGs, of which 257 overlapped with WGCNA-selected genes. These genes exhibited significant enrichment in immune-related pathways, including "cytokine-cytokine receptor interaction" and "chemokine signaling pathway." Gene Set Enrichment Analysis

**Data availability statement:** All relevant data are within the manuscript and its Supporting information files.

**Funding:** This study was financially supported by the Shandong Province Natural Science Foundation Youth Project in the form of a grant (ZR2023QH460) received by HZ. This study was also financially supported by the Key Research and Development Program of Shandong in the form of a grant (2022CXPT023) received by HZ. This study was also financially supported by the Shandong Provincial Technology Innovation Guidance Plan in the form of a grant (YDZX 2023041) received by XL. The funders were involved in specific aspects of the study, including manuscript proofreading and statistical analysis. The funders had no additional undeclared role in study design, data collection, or decision to publish the manuscript.

**Competing interests:** The authors have declared that no competing interests exist.

(GSEA) further uncovered 12 KEGG pathways significantly associated with disease risk scores. Drug screening identified 24 small molecule drugs related to key genes. A diagnostic model incorporating five genes (RFC4, LYN, IL3, TNFRSF1B, and RBBP7) was constructed, demonstrating diagnostic efficiencies of 0.843 and 0.739 in the training and validation sets, respectively. An AR mouse model was successfully established, and the expression levels of relevant genes were validated through RT-qPCR experiments. The five-gene diagnostic model established in this study exhibits strong predictive ability in distinguishing AR patients from healthy controls, with potential clinical applications in diagnosing AR and advancing novel diagnostic and therapeutic strategies.

## 1. Introduction

Allergic Rhinitis (AR) is a prevalent chronic inflammation of the nasal mucosa triggered by allergic reactions, making it one of the most common diseases globally [1,2]. The self-reported prevalence of AR varies significantly, with estimates ranging from 2% to 25% in children and 1% to 40% in adults, exhibiting an increasing trend over time [3,4]. AR manifests with symptoms including nasal congestion, runny nose, sneezing, and itching due to exposure to inhaled allergens, leading to inflammation of the mucous membranes [5].

CD4 + T cells, which serves as pivotal immune cells within the human immune system, play a central role in modulating various immune responses both directly and indirectly, such as the secretion of interleukin-10 and the inhibition of target cell activation [6,7]. Recent studies have highlighted the involvement of CD4 + T cells in the pathophysiology of AR [8]. Upon exposure to allergens in the nasal cavity, CD4 + T cells become activated and differentiate into distinct T helper cell subsets, including Th1, Th2, and Treg cells [9–12]. In AR, the activation and exaggerated response of Th2 cells are considered key contributors to disease progression [8]. In this study, microarray data derived from blood CD4 + T cell samples related to AR were systematically analyzed. Our aim was to construct a diagnostic model based on characteristic gene signatures of these samples and to identify potential therapeutic targets by evaluating the molecular profiles of immune cells and integrating their gene expression features.

## 2. Materials & methods

### 2.1 Data set acquisition and batch correction

Two datasets were downloaded from NCBI GEO database [13] for analysis: GSE5010 [14], which consists of 38 human blood-extracted CD4 + T cell samples (18 AR [Allergic Rhinitis] patients and 20 from CTRL [Control] subjects), with the detection platform being the GPL10558 Illumina HumanHT-12 V4.0 expression beadchip; and GSE50223, which contains 84 human blood CD4 + T cell samples (42 from AR patients and 42 from CTRL subjects), with the detection platform being the GPL6884 Illumina HumanWG-6 v3.0 expression beadchip. Given that these datasets originate

from different batches of gene expression data, batch effect removal was applied using the sva package (Version 3.38.0) [15] in R (Version 4.3.1) to mitigate potential batch effects. Subsequently, the datasets were integrated to generate combined gene expression profiles for subsequent analysis.

## 2.2 Assessment of sample immune cell proportion

To assess the types of immunoinfiltration in the combined samples, immunologic signature gene sets were obtained from the GSEA database [16]. Subsequently, the GSVA package (Version 1.36.3) [17] in R (Version 4.3.1) was utilized to evaluate the immunoinfiltration profiles across the combined samples. Additionally, the estimate package in R (Version 4.3.1) [18] was employed to calculate immunity scores, matrix scores, and ESTIMATE scores for all samples. Thereafter, Kruskal-Wallis tests were conducted to compare the proportion of immune cells between AR and CTRL samples and to ascertain significant differences in the ESTIMATE scores.

## 2.3 Screening genes associated with diseases based on WGCNA

WGCNA algorithm, implemented through the WGCNA package (Version 1.61) [19] in R (Version 4.3.1), was employed to construct gene co-expression networks and identify modules associated with the disease state in the combined dataset. The implementation of the WGCNA algorithm involves several key steps, including the definition of the adjacency function and the partitioning of modules. For module partitioning, the following thresholds were applied: each module within the module set contains at least 100 genes, and the cutHeight parameter was set to 0.995. Modules with an absolute correlation with disease status higher than 0.3 were retained, along with their constituent genes, as candidate genes associated with AR. Through adherence to these parameters and thresholds, robust gene modules strongly correlated with the disease phenotype were identified, thereby facilitating the exploration of potential pathogenic mechanisms and therapeutic targets related to AR.

## 2.4 Identification of DEGs

In the combined dataset, samples were divided into AR and CTRL groups based on their origin. Subsequently, R (version 4.3.1) and the limma software package (version 3.34.7) [20] were employed to identify DEGs between the two groups (P ≤ 0.05, fold change≥1.2). The identified DEGs were then compared with candidate genes obtained through WGCNA analysis, and the overlapping genes were retained for further analysis. Finally, DAVID (version 6.8) [21,22] was used to perform GO biological process and KEGG pathway enrichment analyses on the retained overlapping genes, with an enrichment significance threshold set at p-value less than 0.05.

## 2.5 Protein–protein interaction (PPI) network construction and identification of key Genes

The STRING database [23] was utilized to investigate the interaction relationships among the protein products of the overlapping significant DEGs identified in the previous step, thereby constructing a protein-protein interaction (PPI) network. The constructed network was subsequently visualized using Cytoscape (Version 3.9.0) [24]. Subsequently, the cytoHubba plugin (Version 0.1) within Cytoscape 3.9.0 [25] was employed to identify key genes in the PPI network based on four topological analysis algorithms: Maximum Clique Centrality (MCC), Neighborhood Component Centrality (MNC), Degree Centrality (DEGREE), and Edge Percolated Component (EPC). Finally, the key genes obtained from each algorithm were compared, and the overlapping genes identified by all four algorithms were retained as the final set of candidate genes.

## 2.6 Construction of a disease diagnosis model

The merged sample dataset was utilized as the training set for regression analysis performed using the lars package (Version 1.2) [26] in R (Version 4.3.1), based on the expression levels of the important candidate genes identified in the

previous step. Subsequently, feature gene selection was conducted using the Support Vector Machine (SVM) method [27] from the e1071 package (Version 1.6−8) in R (Version 4.3.1) to construct a disease diagnosis classifier based on the selected feature genes. The model's discriminatory ability was then evaluated using the ROC curve method implemented via the pROC package [28] in R (Version 4.3.1). To evaluate the model's disease discrimination capability, the AR-related dataset GSE43523 was retrieved from the NCBI GEO database. Additionally, a Nomogram model was constructed using the rms package (Version 5.1−2) [29] in R (Version 4.3.1). Decision curve analysis for both individual and multiple gene combination models was conducted using the rmda package (Version 1.6) [30] in R (Version 4.3.1) to assess the net benefit rate of each gene's effect on disease outcome, thus comparing the impact of different genes on sample classification. In the validation dataset GSE43523, a Nomogram model was similarly constructed based on the previously selected DEGs factors to validate the diagnostic model's efficacy.

## 2.7 Gene set enrichment analysis related to risk scores

Based on the expression levels of all detected genes in the samples from the merged dataset, GSEA [31] was utilized to identify KEGG signaling pathways significantly associated with the risk scores of disease samples. A threshold of False Discovery Rate (FDR) less than 0.05 was applied to determine the significantly enriched KEGG pathways.

## 2.8 Screening of important gene-related drugs

In the Comparative Toxicogenomics Database 2023 update [32], a systematic search was conducted using 'Allergic Rhinitis' as the key term to identify all factors related to the disease (including drugs, environmental factors, etc.). Among these factors, small molecule drugs that exhibited associations with the genes utilized for constructing diagnostic models were selected, thereby establishing connections between diagnostic genes and disease etiological factors. This process facilitated the construction of relationships between diagnostic genes and disease causative factors.

## 2.9 Mice

Female BALB/C mice, aged six weeks, were acquired from the AMDOC company and maintained under specific pathogen-free (SPF) conditions at the Yuhuangding Hospital affiliated with Qingdao University. Prior to the experiment, the mice were acclimatized in an isolation room for one week. Emphasis was placed on maximizing animal welfare and minimizing distress in accordance with ethical guidelines for laboratory animal care.

## 2.10 Establishment of the mouse AR model

According to the established protocol, mice were randomly divided into two groups: the AR group and the control group, with six mice in each group. For the AR group, basic sensitization was performed every other day from days 0–14 using a 1 ml sterile syringe to inject 200ul of saline containing 25ug of ovalbumin (OVA) and 2 mg of aluminum hydroxide (Al(OH)$_3$) intraperitoneally. Mice in the control group received an equivalent volume of physiological saline at the same time intervals. From days 15–21, mice in the AR group were intranasally challenged daily with a 5% OVA solution (20ul per nostril), while mice in the control group were administrated an equal volume of physiological saline intranasally at the same time each day. Half an hour after the nasal challenge, behavioral observation, including and recording of nasal discharge, nose scratching, and sneezing were recorded every 5 minutes. All mice were euthanized within 24 hours following the final nasal challenge procedure. Euthanasia was conducted under isoflurane anesthesia via rapid cervical dislocation to minimize suffering, with death confirmed by the absence of corneal reflex and cardiorespiratory arrest. Subsequent experiments materials were collected for further analysis. All animal experiments were approved by the Animal Experiment Ethics Committee of Yuhuangding Hospital (Approval NO: 2025−083).

## 2.11 Collection of laboratory animal tissues and blood samples

Within 24 hours following the final treatment, blood was collected from the mice via enucleation after euthanasia and allowed to clot at room temperature for approximately 2 hours. Subsequently, serum was separated by centrifuging the blood at 860×g for 10 minutes at 4°C and stored at −80°C for further analysis. A portion of the nasal mucosa tissue from three mice per group was stored in a −80°C freezer for subsequent experiments. Meanwhile, the remaining mouse heads were fixed in a 4% paraformaldehyde solution and stored at room temperature for histological section preparation and staining.

## 2.12 Histopathological analyses of nasal tissues

Mouse heads preserved in 4% paraformaldehyde solution were subjected to decalcification in EDTA buffer for two weeks, followed by paraffin embedding and sectioning. The sections were subsequently deparaffinized by soaking twice in xylene for 20 minutes each, then rehydrated through a graded ethanol series: immersion in absolute ethanol for 5 minutes, followed by 75% alcohol for 5 minutes, and finally rinsed with distilled water. Hematoxylin and eosin (HE) staining was performed according to the standard protocol: sections were stained in hematoxylin solution for 5 minutes, rinsed with water, differentiated in a differentiation solution, and washed again with water. Subsequently, the sections were dehydrated through graded alcohol solutions (85% and 95% alcohol for 5 minutes each), stained with eosin for 5 minutes, dehydrated in absolute ethanol for 5 minutes, and cleared in xylene for 5 minutes. (two changes). Finally, the sections were mounted with a coverslip for microscopic examination. Observations were conducted under an optical microscope, and images were captured and analyzed.

## 2.13 Measurement of IgE and inflammatory cytokines in serum

Serum levels of IgE (Beyotime, PI476), IL-4 (Beyotime, PI612), IL-5 (Beyotime, PI620) and IL-13 (Beyotime, PI539) in the control and AR groups were measured using ELISA kits, according to the manufacturer's instructions. The absorbance was measured at 450 nm using a microplate reader, and the mean values were calculated based on triplicate measurement.

## 2.14 RT-qPCR

To investigate the expression of relevant genes in a mouse model of AR using RT-qPCR, five nasal mucosa samples were collected from AR-model mice and five samples from normal mice. Total RNA was extracted from the nasal mucosa of AR-model mice using TRIzol Reagent (Beyotime, China). Subsequently, the quality and concentration of the RNA were measured. The extracted RNA was reverse-transcribed into cDNA. RT-qPCR analysis was performed with GAPDH as an internal reference gene. The detailed primer sequences are listed in Table 1, and differential gene expression was quantified using the $2-\Delta\Delta Ct$ method.

# 3 Results

## 3.1 Assessment of immunocyte proportions in samples

The relationships among samples before and after batch effect removal using the sva algorithm were analyzed and are presented Fig 1. The expression profile data are provided in S1 File and S2 File. The proportions of 28 immune cell types, as depicted in Figs 2A and 2C, are detailed in S3 File. Comparative analysis of immune cell proportions between the AR and CTRL groups revealed significant differences in 8 immune cell types. The p-values for statistical tests of each immune cell type are also included in S3 File. Additionally, significant differences in score estimates were observed across all samples between the AR and CTRL groups, as shown in Fig 2B and documented in S4 File.

**Table 1. Correlated primer sequence table.**

| Primer Name | Primer Sequence |
| --- | --- |
| TNFRSF1B-Forward | 5′-TCCTGGCTATTCCCGGAAATG-3′ |
| TNFRSF1B-Reverse | 5′-TGTAAGGATGCTTGGAGTTTGG-3′ |
| RFC4-Forward | 5′-ACAAGTAGTCCGAGAGAAAGTGA-3′ |
| RFC4-Reverse | 5′-CTTAAAGGGAGGACATGGCTTC-3′ |
| RBBP7-Forward | 5′-GAGCGTGTCATCAACGAAGAG-3′ |
| RBBP7-Reverse | 5′-GCATGGGTCATAACCAGGTCATA-3′ |
| IL3-Forward | 5′-GGGATACCCACCGTTTAACCA-3′ |
| IL3-Reverse | 5′-AGGTTTACTCTCCGAAAGCTCTT-3′ |
| Lyn-Forward | 5′-CATCTCTCCTCGCATCACTTT-3′ |
| Lyn-Reverse | 5′-GGATCTCCCAGGCATCTTTATC −3′ |
| GAPDH-Forward | 5′-GTCTCCTCTGACTTCAACAGCG-3′ |
| GAPDH-Reverse | 5′-ACCACCCTGTTGCTGTAGCCAA-3′ |

## 3.2 WGCNA

In Fig 3A, a power value of 9 was chosen when the square of the correlation coefficient reached 0.9, yielding an average node connectivity of 1 in the constructed co-expression network, which is consistent with the properties of small-world networks. Subsequently, dissimilarity coefficients between gene points were calculated, and a hierarchical clustering tree was generated (Fig 3B). Each module contained a minimum of 100 genes, and a pruning height of cutHeight = 0.995 was applied, resulting in 8 modules with gene details provided in S5 File. Correlations were then computed among sample disease states, 8 immune cell types, 3 estimate scores, and the partitioned modules. Three modules—blue, brown, and turquoise—were retained based on their absolute correlation with disease states exceeding 0.3. These modules, along with their constituent genes (a total of 4278 genes), were identified as candidate genes associated with the disease. Notably, a highly significant correlation was observed between the retained three modules and Activated CD4 T cell, as shown in Fig 4.

## 3.3 Significantly DEGs

The results of differential expression analysis are presented in the form of volcano plot in Fig 5A. This plot provides comprehensive overview of both biological magnitude and statistical significance of DEGs. Thresholds of demarcation are indicated by vertical dashed lines at $|\log_2 FC| = 0.263$ and horizontal dashed line at FDR = 0.05, dividing the scatterplot into three distinct regions: significantly upregulated genes (upper right quadrant, n = 187), significantly downregulated genes (upper left quadrant, n = 139), and non-significant genes (central region). A total of 326 significantly DEGs were identified. This visualization validates the appropriateness of the screening thresholds and highlights target gene sets for downstream functional analyses. The complete list of significantly DEGs is available in S6 File. Comparison of these 326 genes with the 4278 genes identified through WGCNA analysis revealed 257 overlapping genes, as shown in Fig 5B. Detailed information is available in S7 File.

## 3.4 Functional enrichment analysis

The analysis of 257 overlapping significantly DEGs unveiled 47 significantly enriched GO biological processes and 9 KEGG signaling pathways. Detailed data are provided in S8 File. These enrichment results were ranked based on their p-values, with the top 10 presented as depicted in Fig 6. The findings highlight a prominent association between the overlapping DEGs with immune response-related processes, particularly demonstrating significant enrichment within the cytokine – cytokine receptor interaction pathway.

**A**

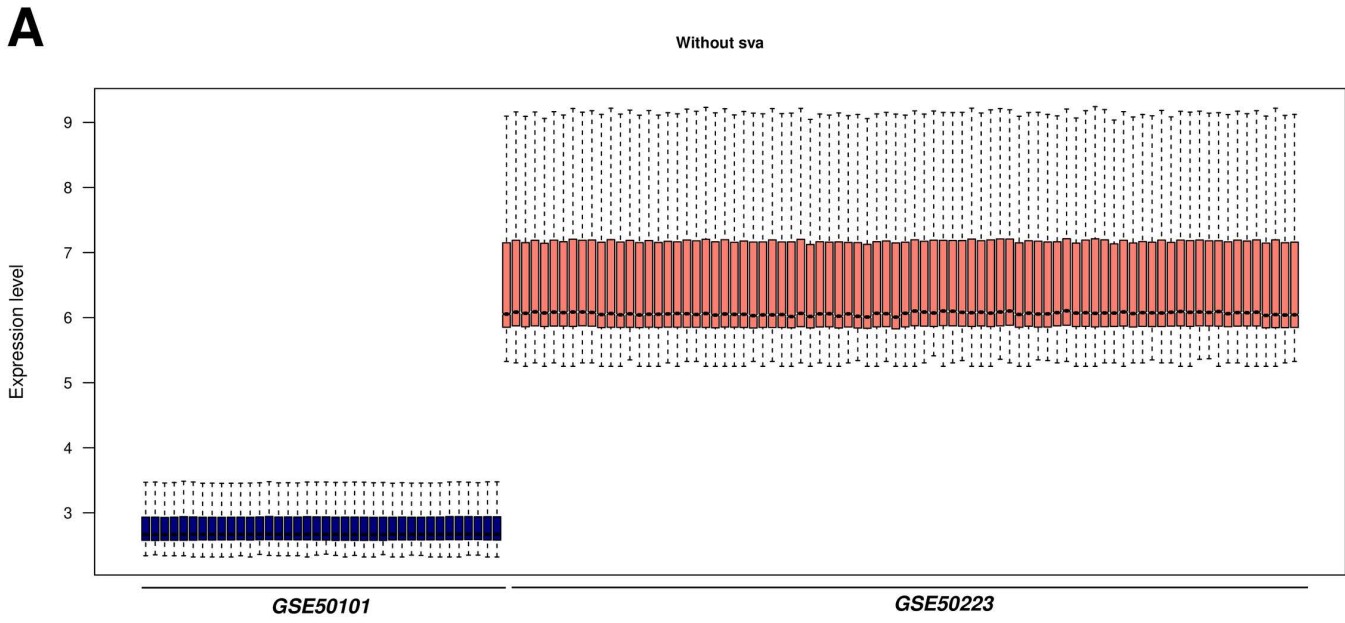

Without sva

**B**

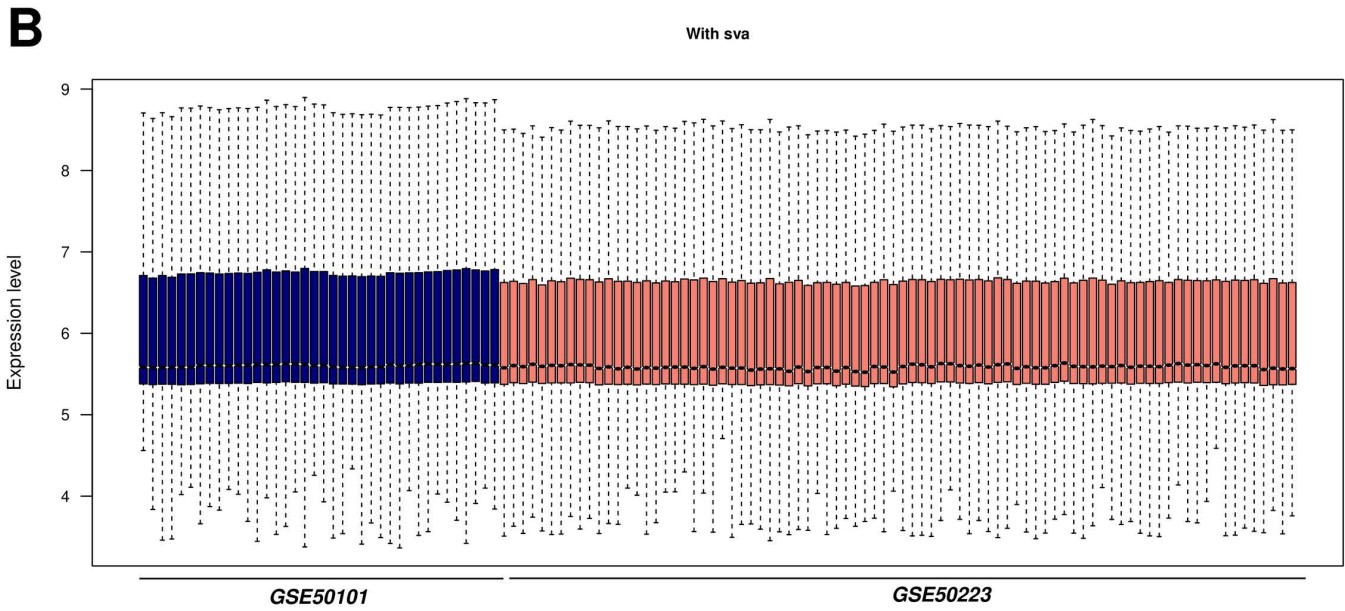

With sva

**Fig 1. sva algorithm processing (A) before and (B) after the sample distribution.**

## 3.5 Construction of PPI networks and identification of important genes

The interaction connections among the protein products of the 257 overlapping significantly DEGs were identified using the STRING dataset. Interactions with a combined score exceeding 0.4 were preserved, resulting in a total of 536 interaction pairs (S9 File). These pairs were utilized to construct an interaction network, as illustrated in Fig 7. A comprehensive search for significant genes was conducted utilizing four topological analysis algorithms: MCC, MNC, DEGREE, and EPC. Detailed results are provided in S10 File. Comparative analysis of the top 20 candidate significant genes from each

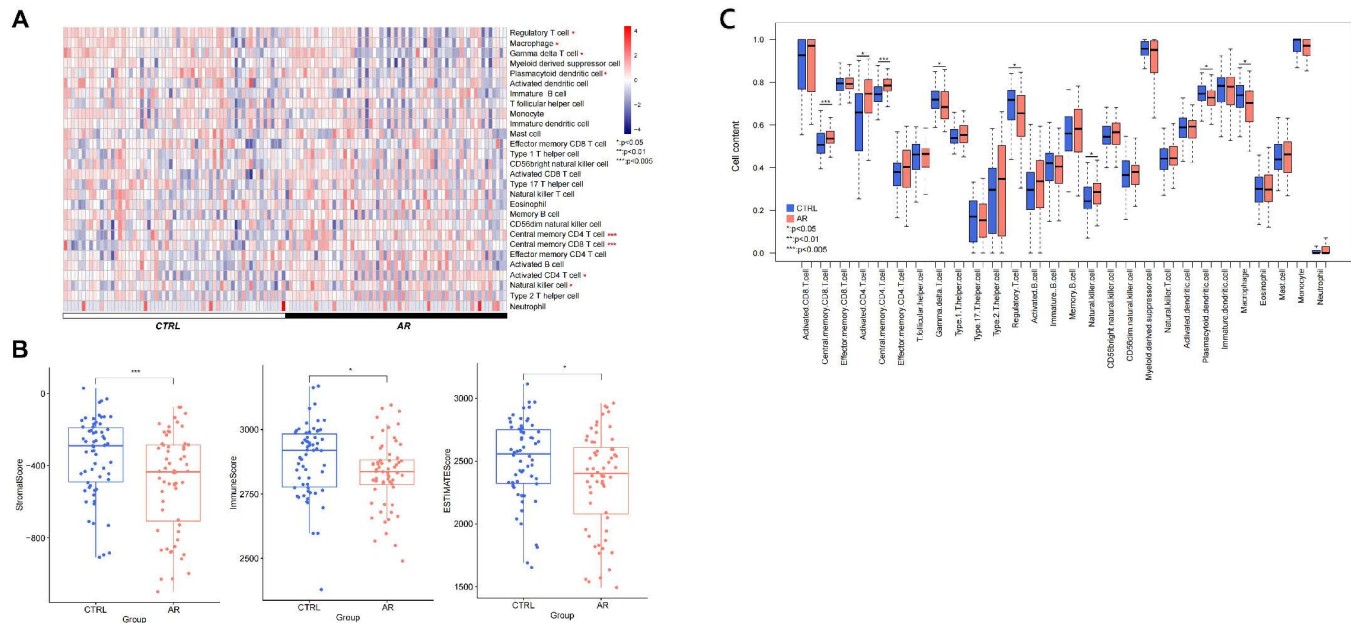

**Fig 2. (A)** Heat map of the distrubution of immune cells in the sample is displayed; **(B)** estimation of each score is displayed; **(C)** sample immune cells distribution box.

algorithm, as depicted in Fig 8, unveiled a total of 14 overlapping genes: IFNG, IL3, IRF4, LYN, NFKBIA, PRF1, RAD54L, RBBP7, RFC4, RRM2, SYK, TIGIT, TNFRSF1B, and TNFRSF4.

### 3.6 Construction of disease diagnosis model

The LASSO regression algorithm was employed to screen for the most optimized gene combinations, as shown in Fig 9. A total of 5 optimal genes were identified: RFC4, LYN, IL3, TNFRSF1B, and RBBP7. The risk coefficients for each gene can be found in S11 File. In the merged training dataset, the expression levels of these five important genes were presented. Simultaneously, the expression trends of these 5 genes between the AR and CTRL sample groups were validated in the validation dataset GSE43523. As depicted in Fig 10, the expression trends of these genes were consistent across both datasets. By utilizing the LASSO coefficients for individual genes (from S11 File), disease risk scores for samples were independently calculated in both the training and validation datasets. As shown in Fig 10, normal control samples demonstrated lower risk scores, whereas disease samples exhibited higher risk scores. The ROC curves for both datasets are illustrated in the right panel of Fig 10. The individual genes exhibited good predictive accuracy within the samples, and their combined effect resulted in the best performance. The diagnostic efficiency of the training set was determined to be 0.843. For the validation set GSE43523, the diagnostic efficiency was found to be 0.739. The nomogram model provided superior clinical utility, as demonstrated in Figs 11 and 12. The expression profile data for the training and validation datasets are presented in S12 File.

### 3.7 KEGG signaling pathway analysis related to risk score

A total of 12 significantly associated KEGG signaling pathways were identified, with the top 5 pathways illustrated in Fig 13. Detailed data are provided in S13 File. The results reveal a notable enrichment of the 'CELL-CYCLE' pathway in the high-risk scoring group, suggesting its potential relevance to disease risk.

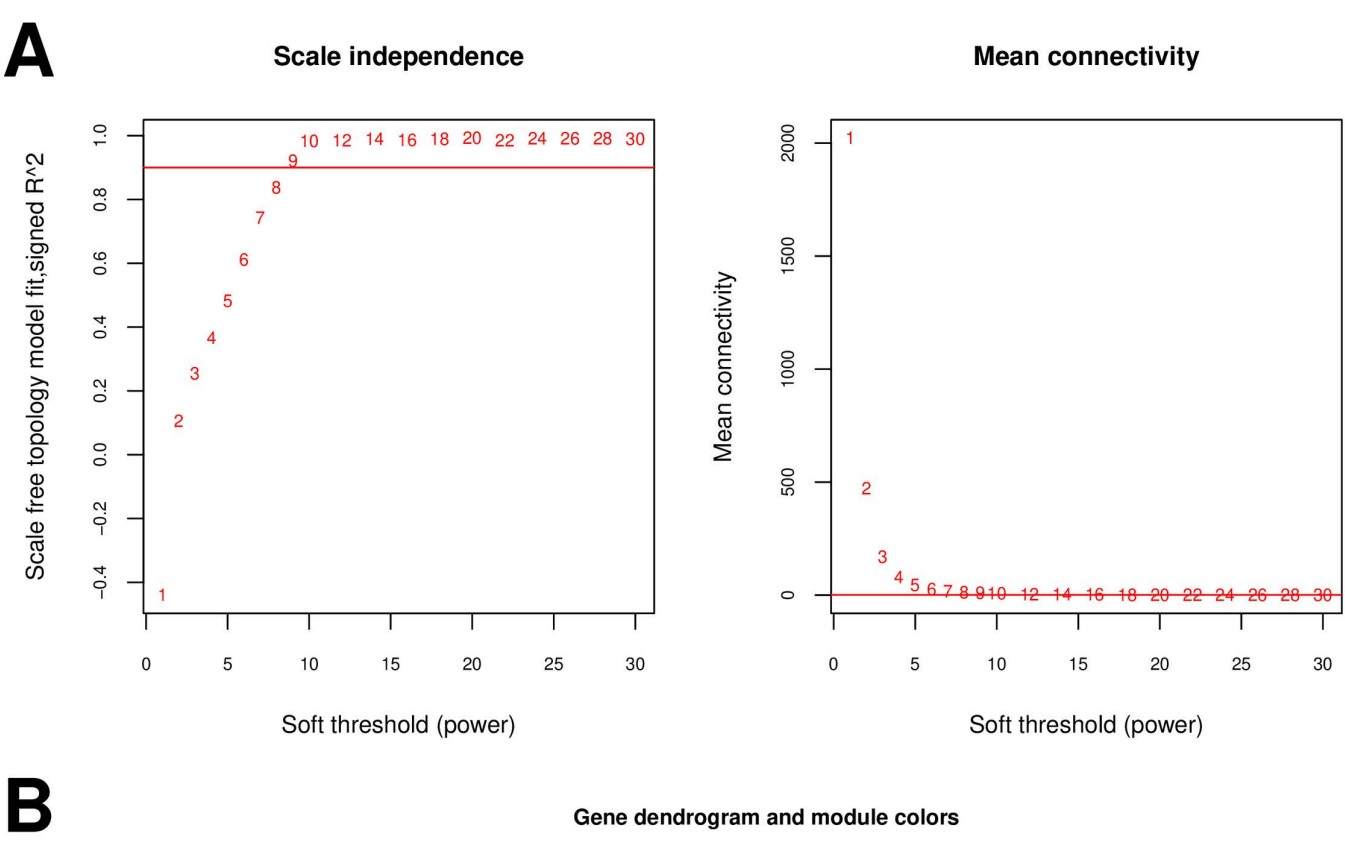

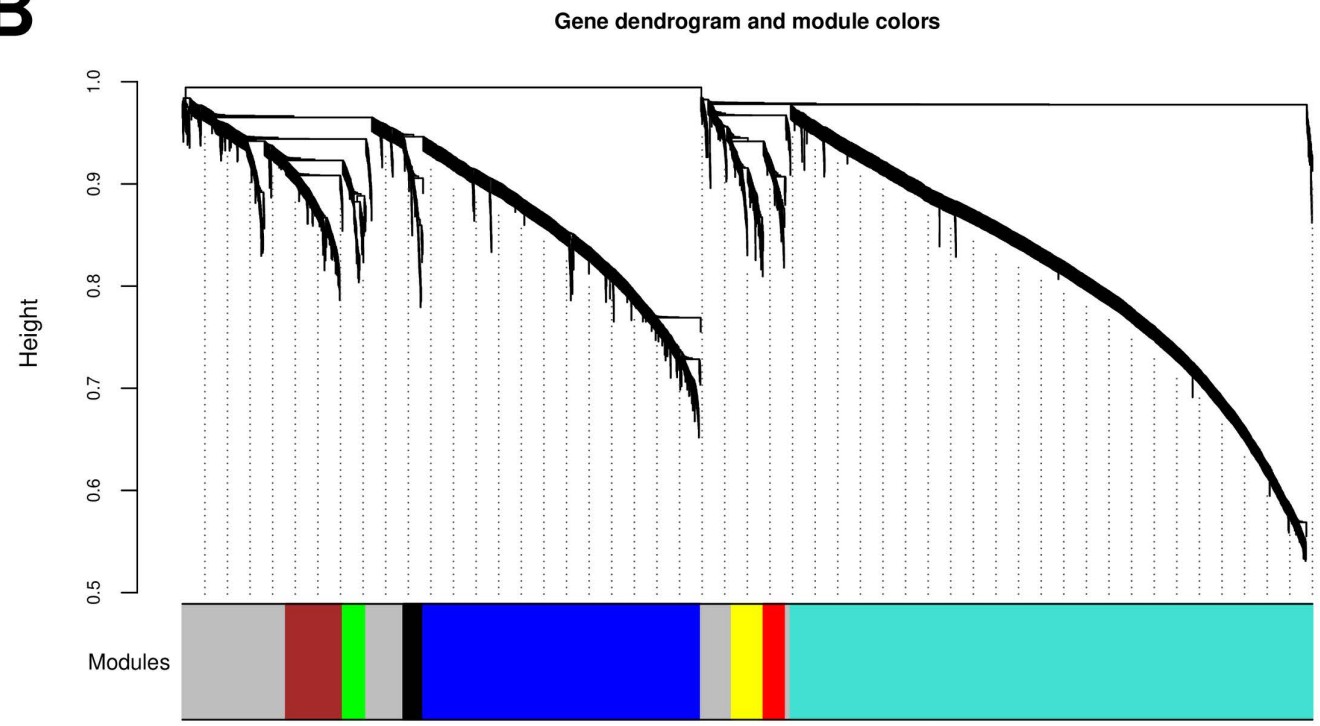

**Fig 3. (A) Left panel: power selection diagram of adjacency matrix weight parameter. Right panel: schematic representation of the average connectivity degree of genes under different power parameters; (B) Module partition tree diagram, each color represents a different module.**

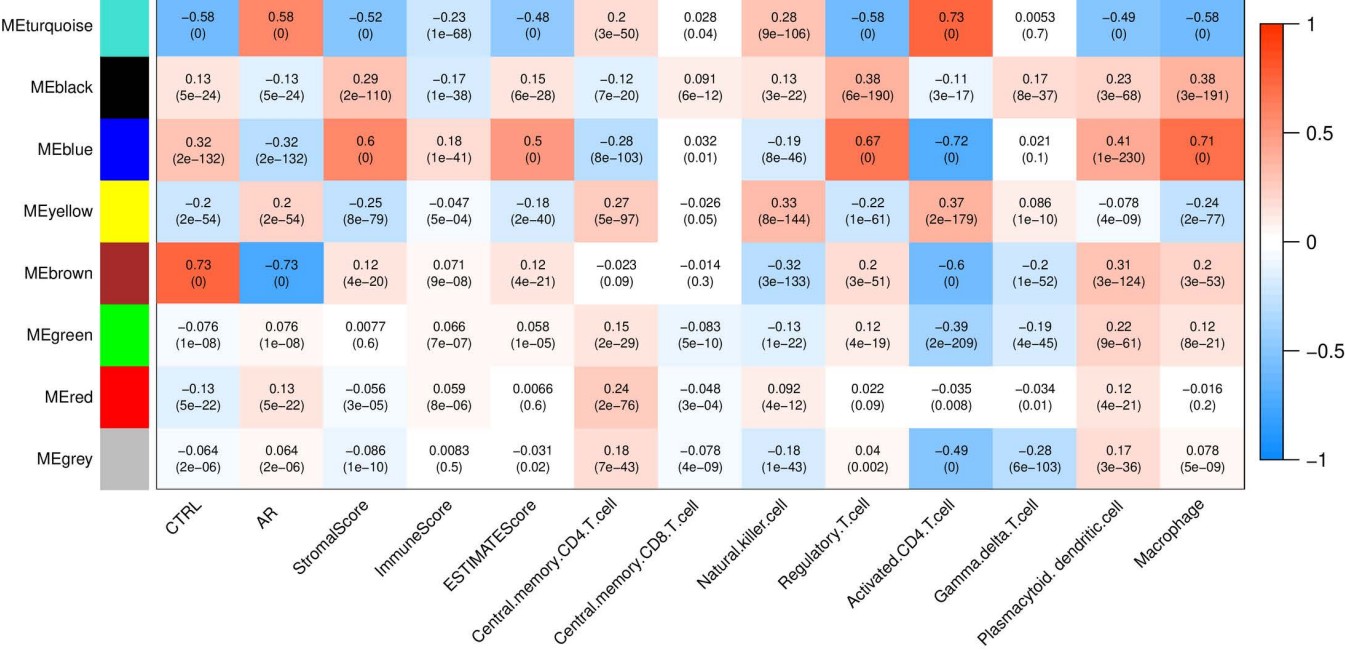

**Fig 4. Module-trait correlation heatmap.**

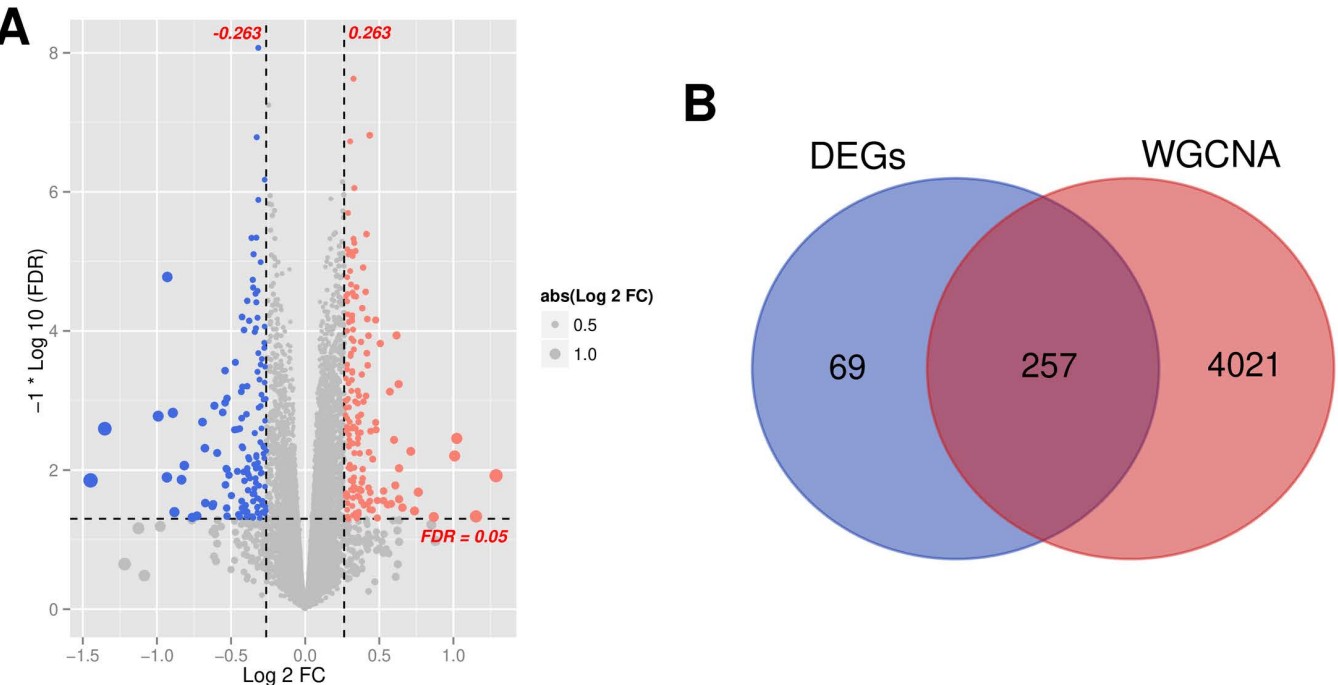

**Fig 5. (A)** Test log2FC-log10 (FDR) volcano plot, blue and red points indicate significantly down-regulated and up-regulated genes, respectively, black horizontal lines indicate FDR < 0.05, and two vertical lines indicate |log2FC| > 0.263; **(B)** Compare Venn diagram.

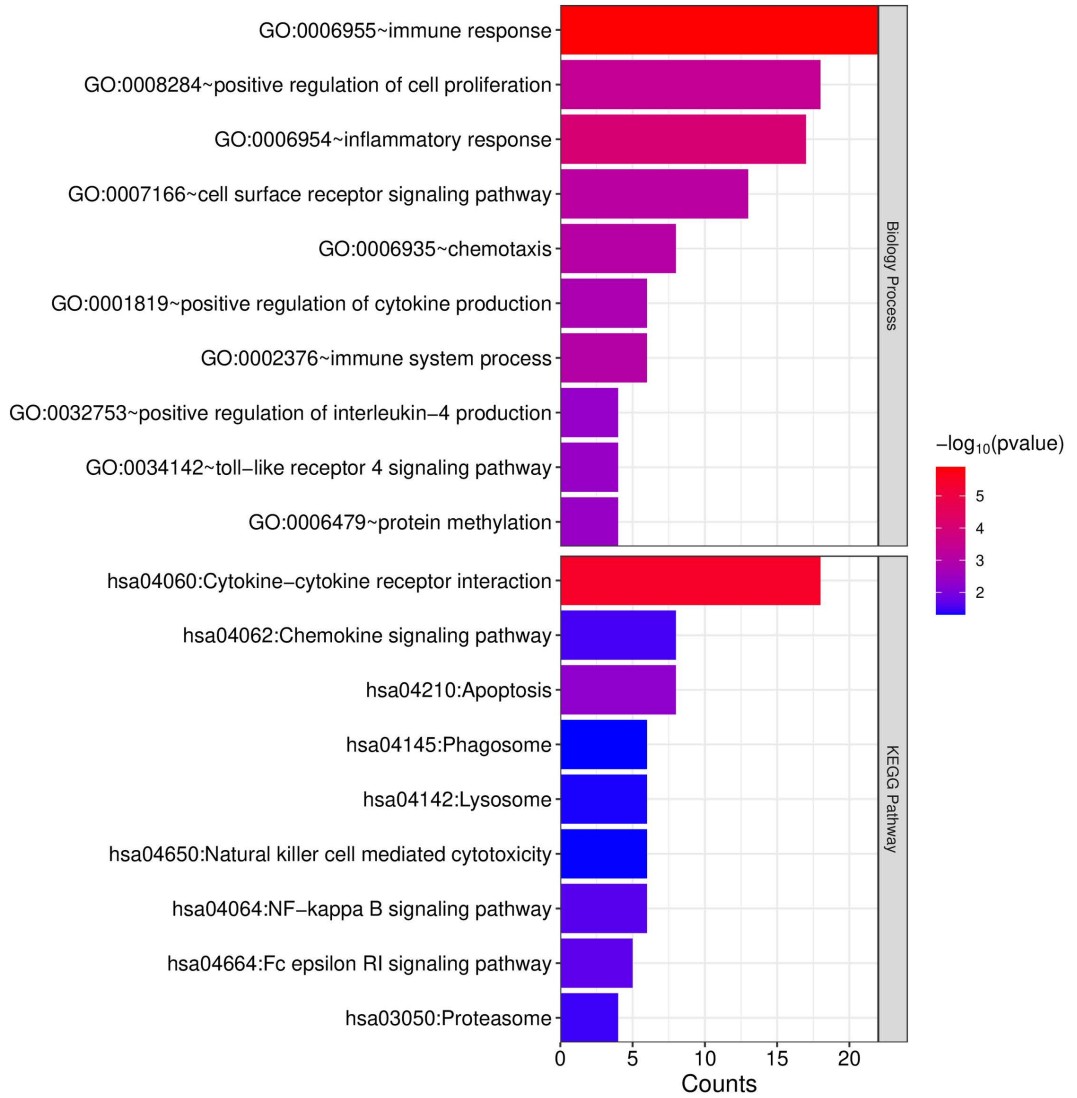

**Fig 6. Enrichment analysis of GO biological processes and KEGG signaling pathways associated with significantly differentially expressed Genes, with the horizontal axis indicating the number of genes, the vertical axis indicating the entry name, and the color indicating the significance.**

### 3.8 Immunocorrelation analysis

In the initial step, the correlation between 8 significantly differentially distributed immune cell types and 3 ESTIMATE scores was calculated, as depicted in Fig 14. Detailed data are available in S14 File. The results reveal that the RFC4 gene exhibits the highest positive correlation with Activated CD4 T cells and the highest negative correlation with Macrophages.

### 3.9 Drug screening for key genes

Small pharmacochemical molecules linked to the 5 diagnostic model genes exhibit a total of 24 pairs of linkage relationships (S15 File), as illustrated in Fig 15. These interactions involve 14 disease-related factors, including Beclomethasone,

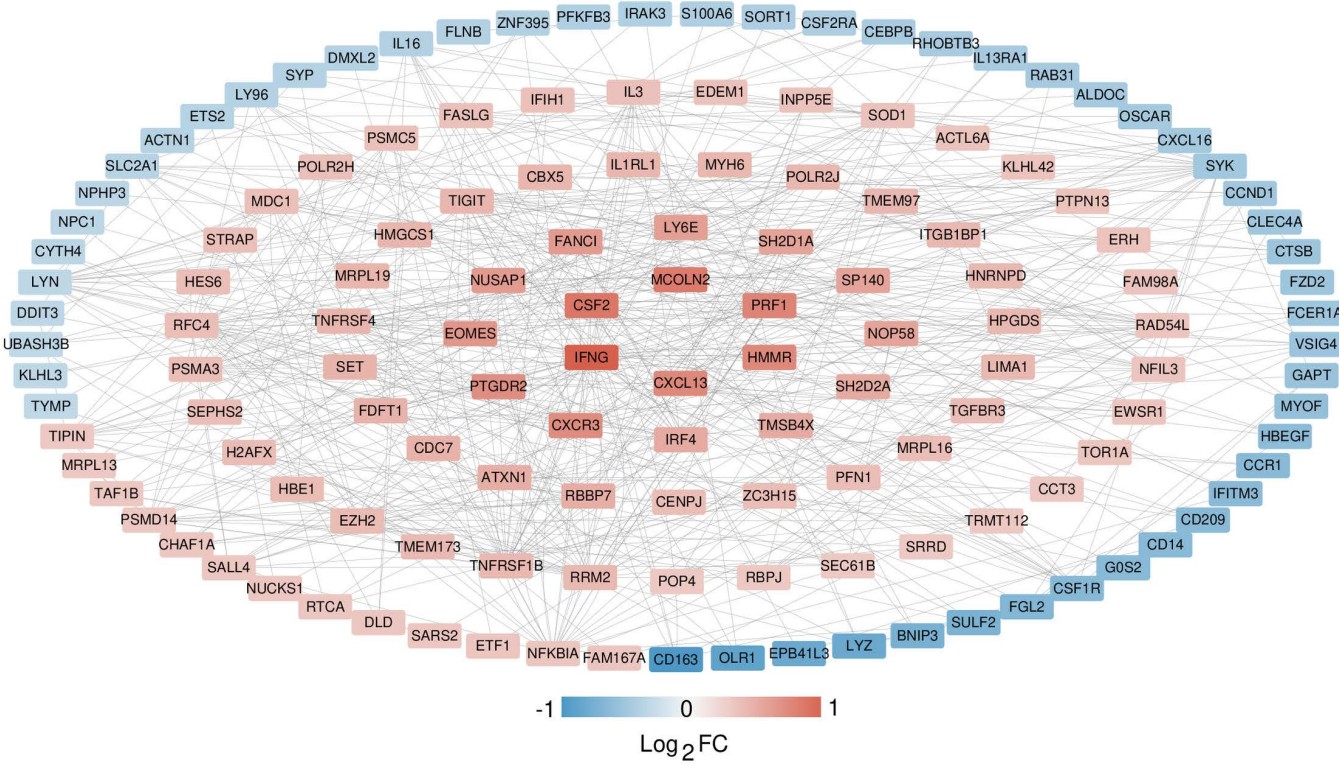

**Fig 7. Interaction network, with color indicating degree of significant difference.**

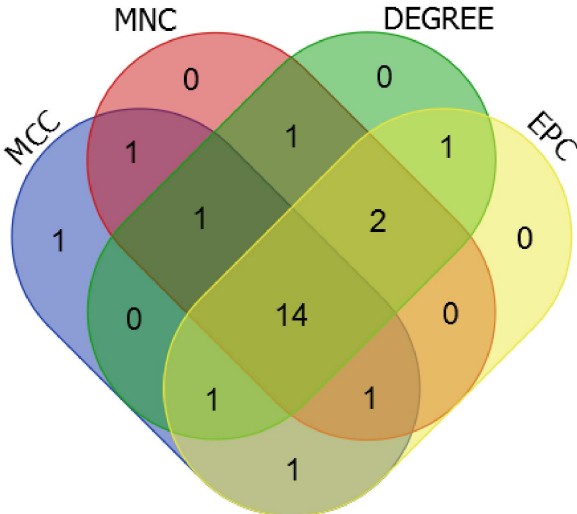

**Fig 8. Venn diagram comparing the top20 candidate important genes of the four algorithms: MCC, MNC, DEGREE and EPC.**

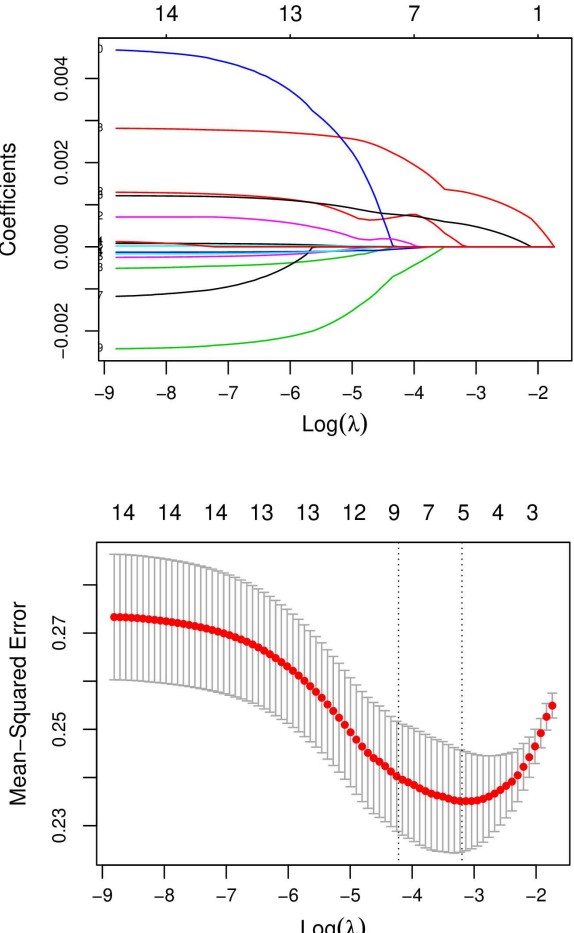

**Fig 9. Parameter map of optimal gene combination screened by Lasso algorithm.**

Benzo(a)pyrene, Methotrexate, Tretinoin, Valproic Acid, Diethylhexyl Phthalate, dorsomorphin, Estradiol, and others. Detailed molecular information is provided in S15 File.

### 3.10 Establishment and evaluation of the AR model

**3.10.1 Observation of general status.** During the experimental observation period, no mortality was observed in either group. Compared with the control group, the total behavioral score of mice in the AR group (Table 2) significantly increased following nasal stimulation. Moreover, the total scores were all greater than or equal to 5 points (Fig 16), indicating that the AR mouse model was successfully established.

**3.10.2 Structural changes of the nasal mucosa.** The staining results indicate that the normal nasal mucosa should exhibit a pseudostratified ciliated columnar epithelium, mainly composed of ciliated columnar epithelial cells and goblet cells, with minimal blood vessels presence, slight eosinophil infiltration, and minor vascular changes (Fig 17A). In the model group, cilia are detached and arranged disorderly, small blood vessels are diffused dilated and the number of mucous glands is increased(Fig 17B).

**3.10.3 Detection of serum IL-4, IL-5, IL-13, and IgE levels.** The levels of IgE, IL-4, and IL-5 in the serum of AR mice were significantly higher than those in control mice (Fig 18).

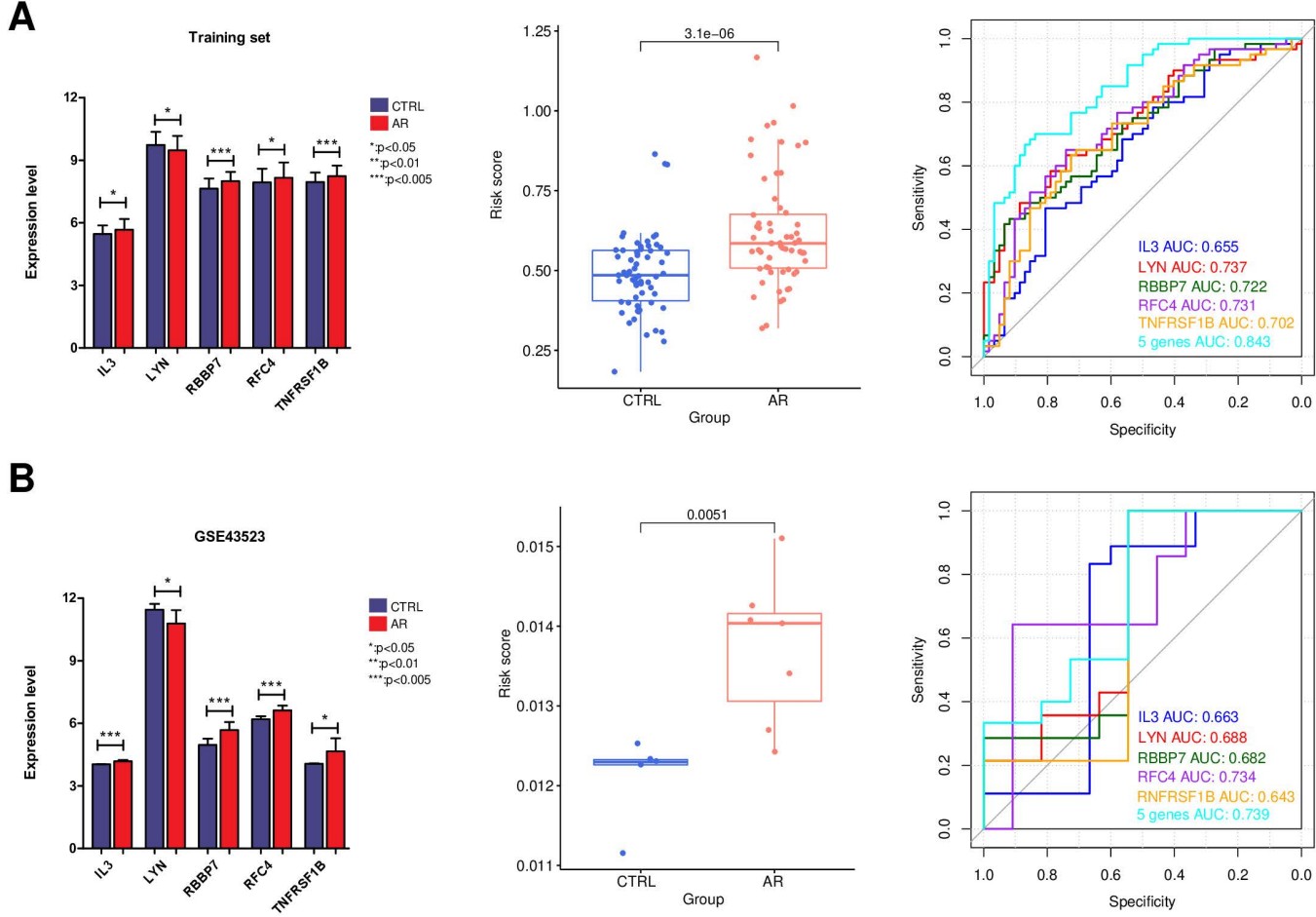

**Fig 10. In training dataset (A) and validation dataset GSE43523 (B), Left panel: expression level distribution map of five important genes in AR and CTRL sample groups; Middle panel: AR and CTRL sample risk score display figure; Right panel: ROC curves of each gene and all genes combined foe sample identification.**

### 3.11 Expression of related genes in AR model mice

The expression of the 5 genes in both groups was further validated using qPCR. Five nasal mucosa samples were collected from AR-model mice and five from normal control mice. As shown in Fig 19, compared with the normal control mice, the expression levels of RFC4, IL3, TNFRSF1B, and RBBP7 genes in the AR group were significantly upregulated. In contrast, the expression level of LYN was downregulated.

## 4 Discussion

AR is an IgE-mediated nasal inflammatory disorder triggered by the introduction of allergens in sensitized individuals [33]. The prevalence of AR varies geographically, ranging from approximately 10% to 40% [34]. Due to global environmental changes and significant genetic predispositions [35], the incidence of AR continues to increase. In severe cases, AR can significantly impair quality of life, disrupt sleep patterns, reduce exercise tolerance, decrease work productivity, and affect social interactions [36]. At present, the diagnosis of AR primarily relies on methods such as the skin prick test (SPT) or in vitro antigen-specific IgE (sIgE) testing. Nevertheless, these diagnostic approaches are susceptible to misdiagnosis and

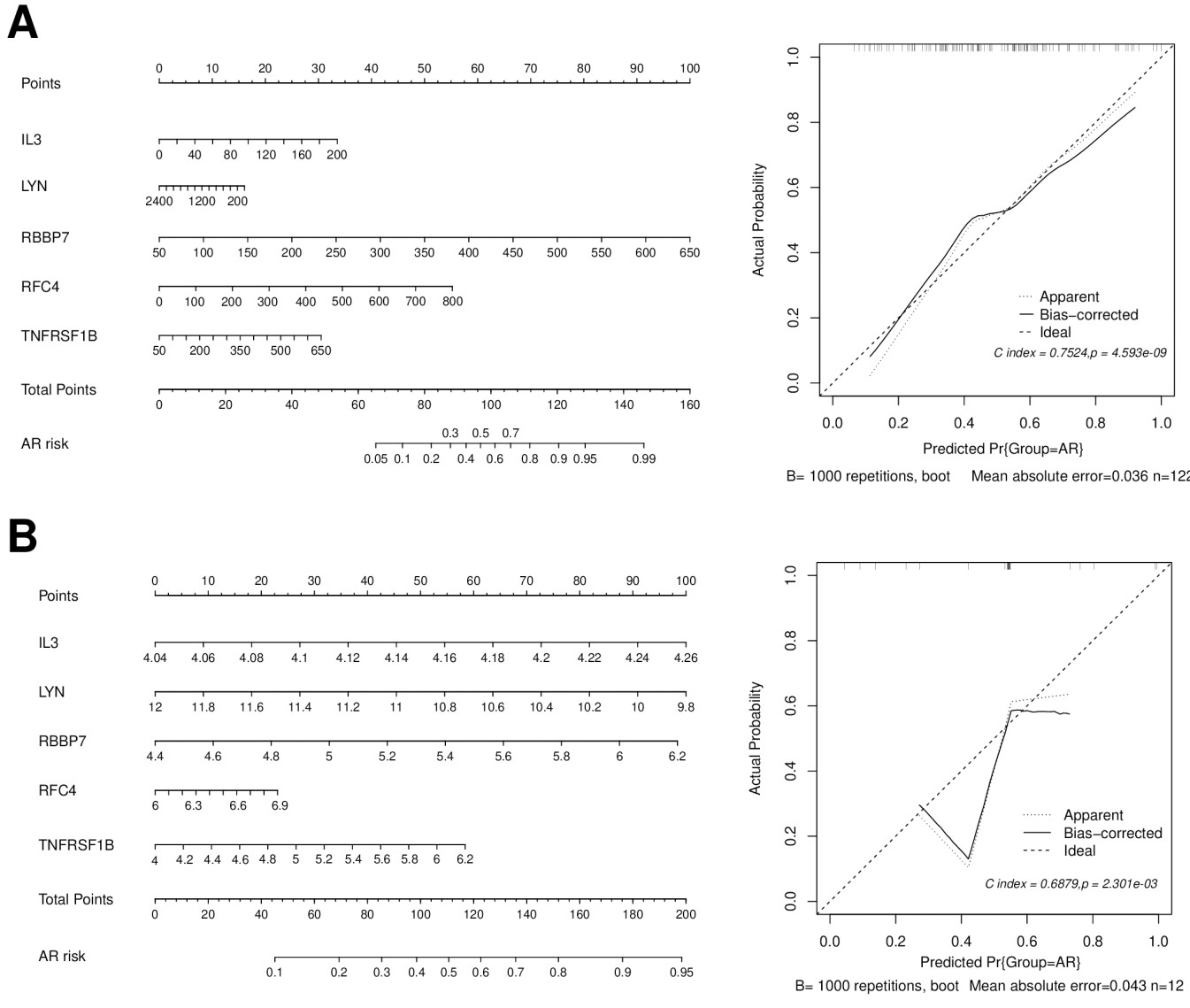

**Fig 11. Presentation of 6SE43523 nomogram and corrected line plot for training dataset (A) and validation dataset (B).**

missed diagnoses due to various influencing factors [37–39]. For instance, in Local Allergic Rhinitis (LAR), which is characterized by localized symptoms and sIgE-mediated inflammatory responses, systemic allergy evidence may be absent, leading to negative test results [38,40,41].

In the present study, immune cell types in each sample were classified using ssGSEA. The proportions of various immune cells were subsequently compared between the AR and CTRL groups, resulting in the identification of eight immune cell types with significant differences. A highly significant correlation with activated CD4 T cells was observed across all three modules retained by WGCNA. Furthermore, five optimized genes were selected, and a predictive model for AR was established. Subsequently, immunocorrelation analysis was conducted, and small molecule drugs were screened. These biomarkers demonstrate potential for clinical applications in the diagnosis of AR, and this study contributes to the advancement of novel clinical diagnostic and therapeutic approaches for this condition.

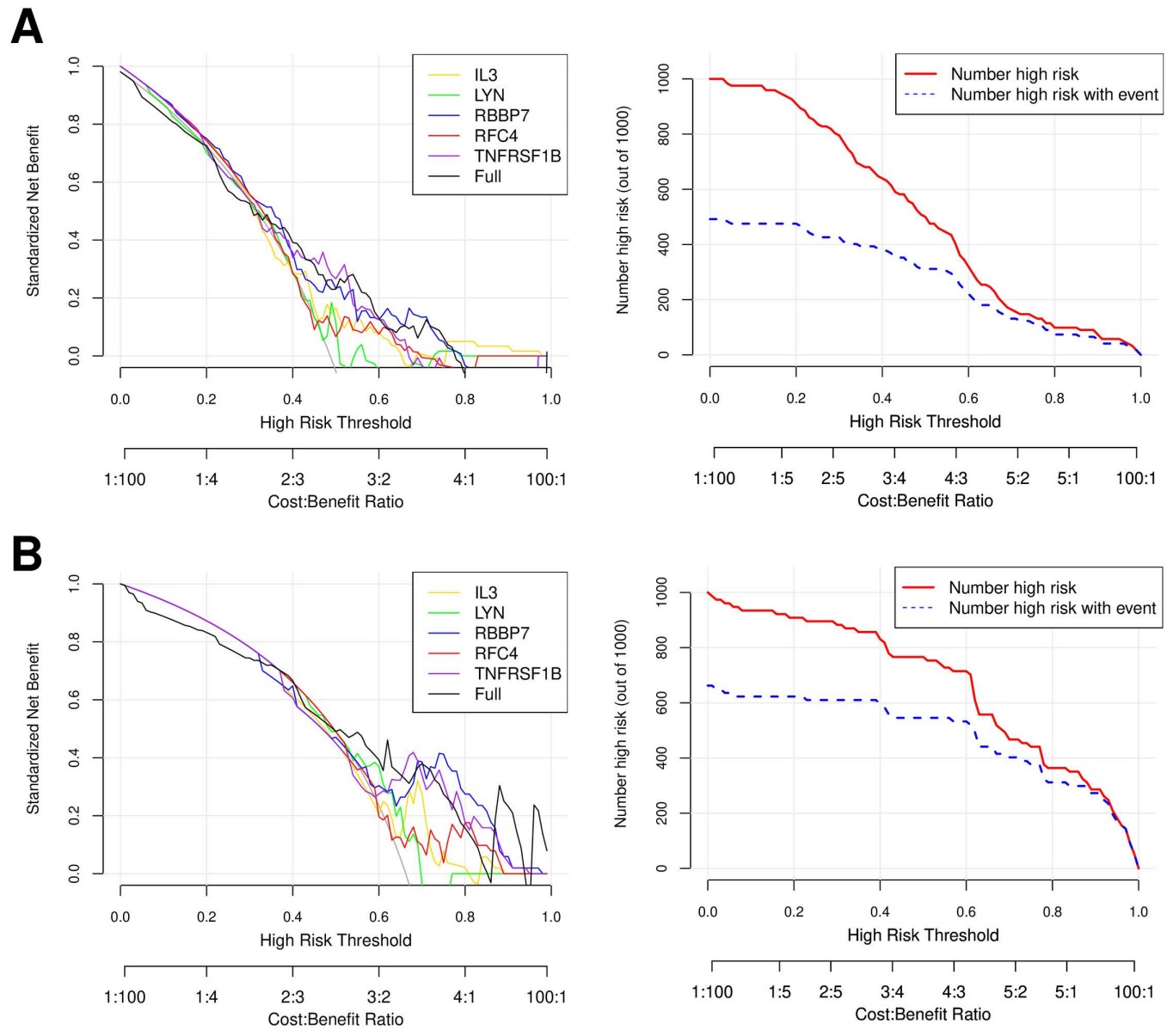

**Fig 12. Decision curves of training dataset (A) and validation dataset GSE43523 (B).**

Increasing evidence suggests that the pathogenesis of AR entails the intricate interplay of diverse immune cells and inflammatory mediators, ultimately leading to nasal mucosal inflammation and the characteristic symptoms of AR [42]. The immune mechanism underlying IgE-mediated AR resembles that of other atopic diseases, involving activation of the adaptive immune system [12]. In patients with AR, specific IgE antibodies are produced and bind to allergens and receptors on mast cell surfaces. Subsequent exposure to the allergen triggers cross-linking of adjacent IgE molecules, leading to mast cell degranulation and particle release. Pre-existing mediators like histamine rapidly activate sensory nerve endings, inducing itching and sneezing, while promoting local vasodilation and glandular secretion, resulting in nasal congestion and discharge. Newly produced mediators, such as leukotrienes, chemokines, and cytokines, contribute to a delayed

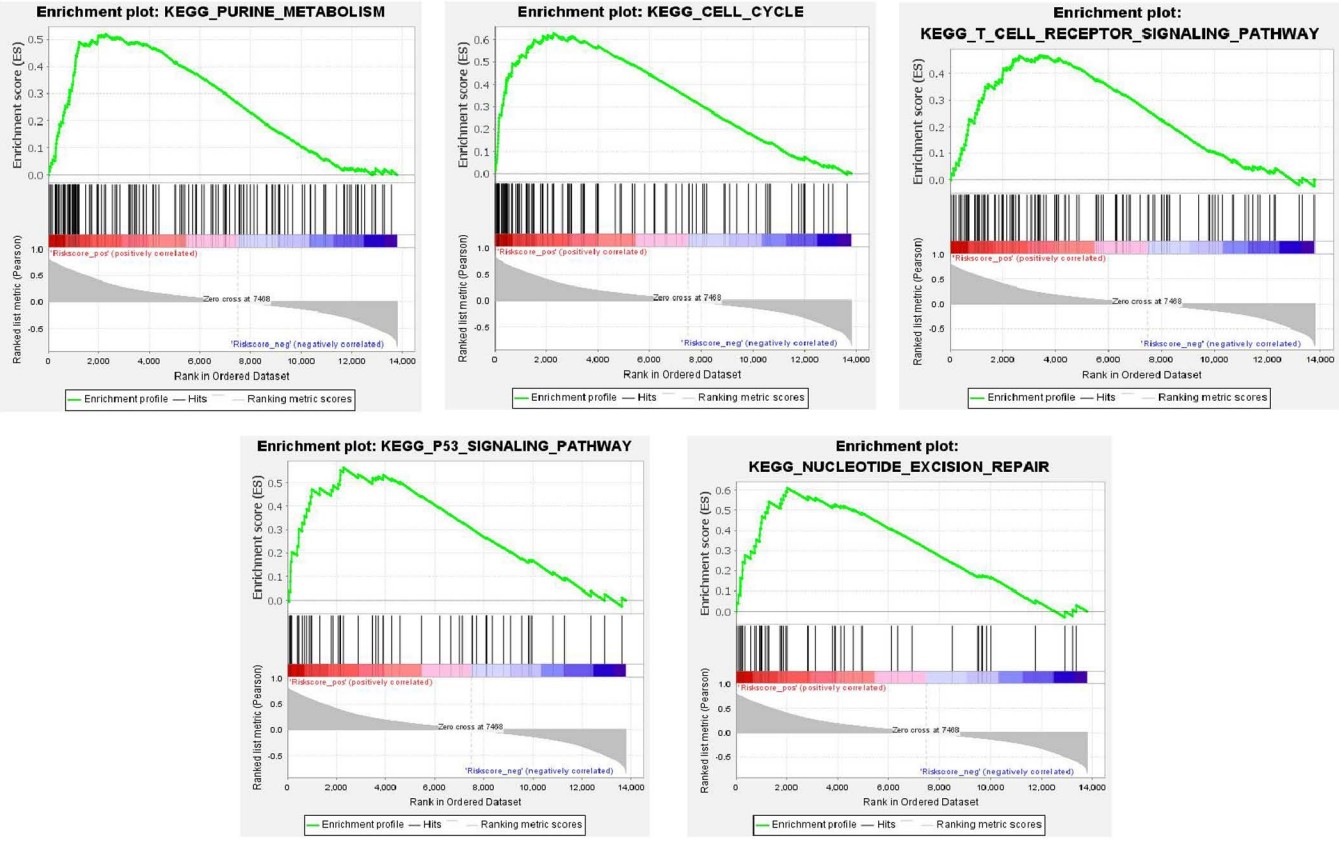

**Fig 13. Risk score significantly correlated KEGG (top 5) signaling pathway.**

inflammatory response mediated by eosinophils and Th2 cells, exacerbating nasal congestion and hyperreactivity [1,2]. Additionally, research has shown that activated CD4 T cells can differentiate into Th2 cells, initiating a Th2-cell-mediated immune response [43]. Consistent with this pathway, our ssGSEA results (Figs 2A and 2C) revealed significantly elevated levels of activated CD4 T cells, central memory CD8 T cells, central memory CD4 T cells, and natural killer cells in the AR group compared to controls. The ESTIMATE package [44], which utilizes gene expression data to predict immune cell infiltration in tumor tissue, was applied to AR for the first time. Our results demonstrated that all immune scores in the AR group were lower than those in the control group, reflecting the low immune status in patients with AR. Currently, there is limited literature accurately describing the immune status of patients with AR, including the proportions and numbers of immune cells in their blood. Additionally, large-scale statistical analysis of immune cell data from clinical blood samples is warranted. GO and KEGG analyses revealed that the overlapping DEGs are primarily involved in biological processes related to 'inflammatory response' and 'immune response'. Additionally, these genes are enriched in pathways such as 'cytokine−cytokine receptor interaction' and 'chemokine signaling pathway', which are directly associated with the pathogenesis of AR.

A five-gene diagnostic model for AR was established. LYN, categorized as a member of the Src family tyrosine kinase [45], mediates critical regulatory functions in immune cells. The signaling pathway involving IgE and its high-affinity receptor (FcεRI) plays a crucial role in regulating allergic reactions. LYN serves as a key signaling molecule that exerts both positive and negative regulatory influences on the IgE/FcεRI signaling pathway, thereby modulating allergic responses. The interaction between LYN and FcεRIβ is indispensable for mast cell activation [46]. Liu et al. investigated an AR mouse

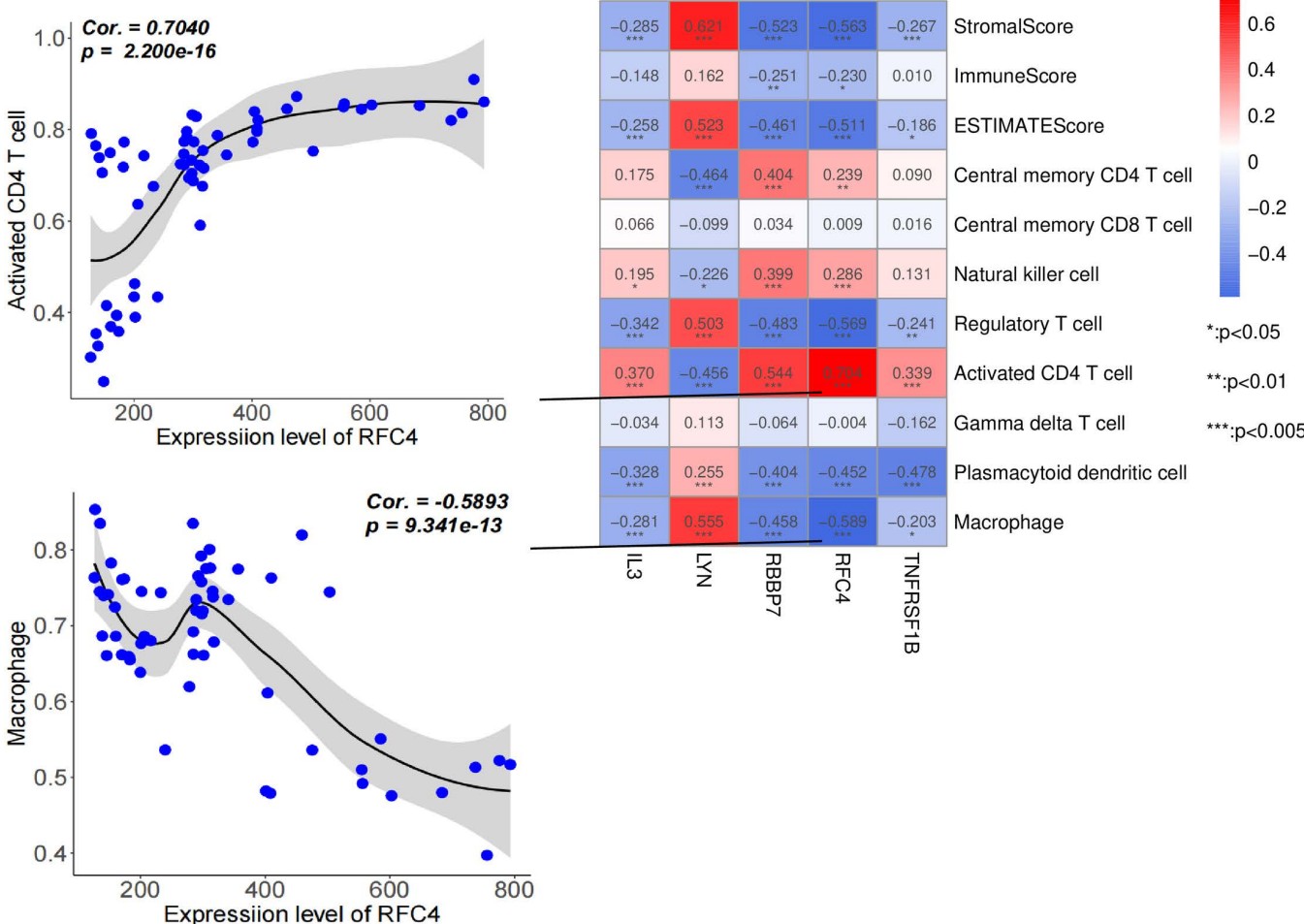

**Fig 14. Presentation of correlations among expression levels of the five model genes and eight significantly differentially distributed immune cells and three estimate scores.**

model, demonstrating that aniline suppressed OVA-induced stimulation while activating IgE-mediated mast cells through a molecular signaling pathway regulated by LYN kinase [47]. IL-3 is predominantly synthesized by various T cell subpopulations [48], although it is also produced by several other immune cells, including basophils, dendritic cells (DC), and mast cells, as well as non-immune cells such as microglia and astrocytes [49]. The activation of basophils and mast cells by IL-3 has been implicated in various chronic inflammatory diseases [50]. However, its specific relationship with AR remains incompletely understood. TNFRSF1B, a member of the tumor necrosis factor receptor superfamily, commonly referred to as TNFR2, plays a crucial role in modulating apoptosis and immune responses [51]. Research indicates that the intensity of TNFR2 signaling can profoundly influence T cell proliferation [52]; however, the precise mechanism underlying its involvement in AR requires further investigation. RFC4, a subunit of the Replication Factor C (RFC) complex, participates in DNA replication and repair processes [53]. RBBP7, a highly conserved WD repeat protein, interacts with histone deacetylases and serves as a component of various co-repressor protein complexes [54]. Research on these two genes has predominantly focused on their roles in tumorigenesis [53,55–57], with limited exploration in the context of allergic diseases. Beclomethasone was identified among the small pharmacochemical molecules linked to the five diagnostic model genes. It inhibits inflammatory cells such as mast cells and eosinophils, thereby reducing the release of mediators

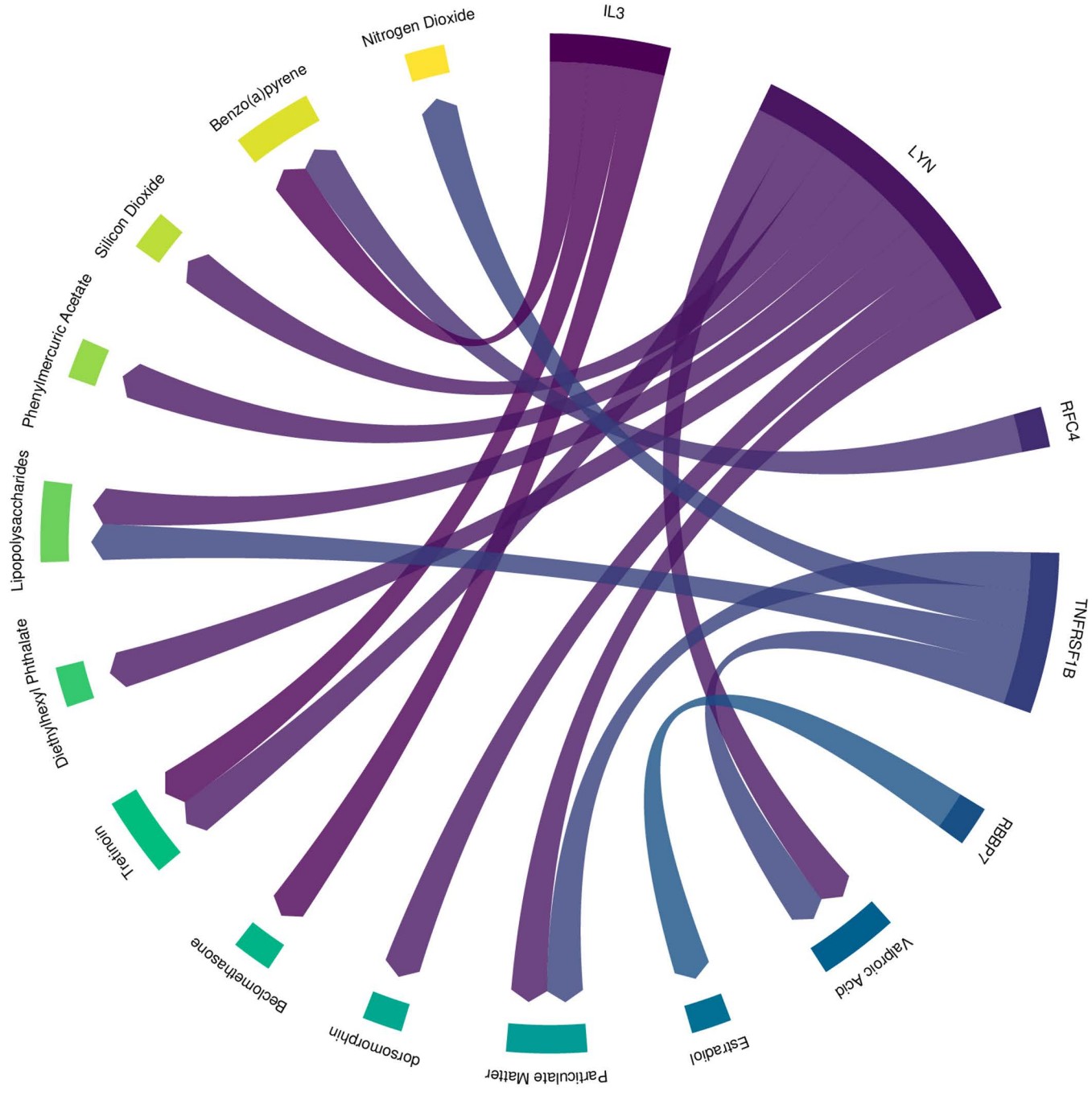

**Fig 15. Diagram of drug relationships associated with Significant genes.**

**Table 2. General symptom scores of the mice.**

| Item | 1 point | 2 points | 3 points |
|---|---|---|---|
| Scratching of the nose | Flicking of the nose several times with one claw | Repeated scratching of the nose with both claws | Friction around the ose |
| Sneezing | 1–3 | 4-10 | >11 |
| Runny nose | Snot flowing into the front nostril | Snot flowing over the front nostril | In tears |

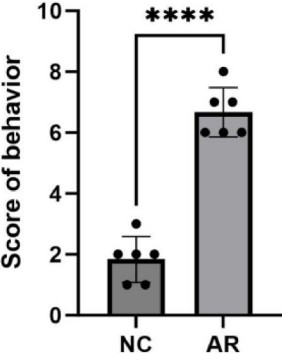

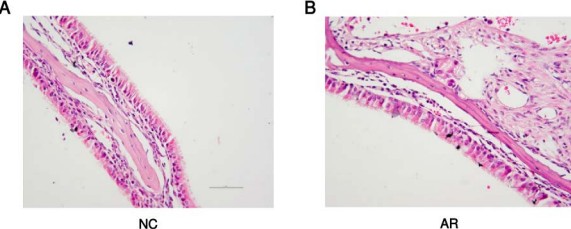

**Fig 16. Behavioral scores of two groups of mice (\*\*\*\*, P<0.0001).**

A

B

NC

AR

**Fig 17. Structural changes of the nasal mucosa.**

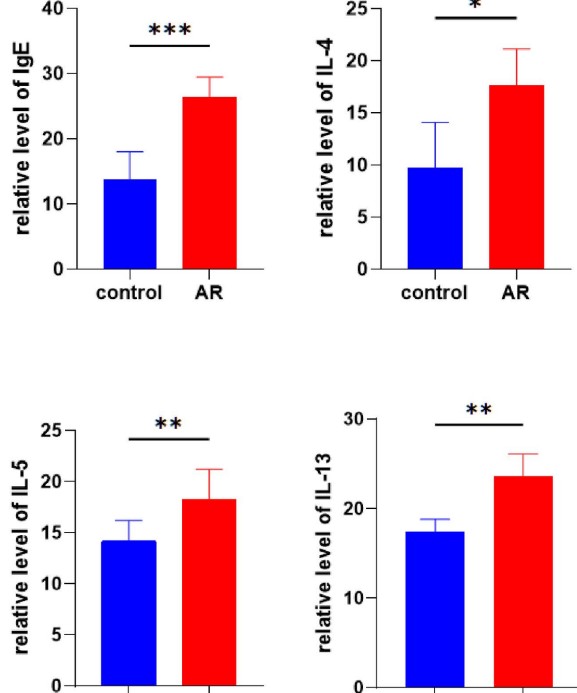

**Fig 18. The expression levels of IgE, IL-4, IL-5, and IL-13 in nasal mucosa of AR and control groups were detected by ELISA.**

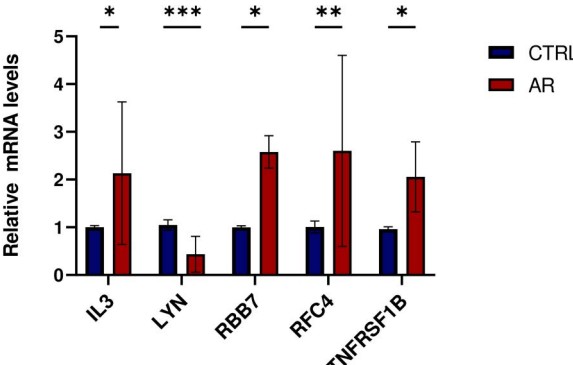

**Fig 19. The expression of IL-3, LYN, RBB7, RCF4, and TNFRSF1B in nasal mucosa of AR and CTRL groups were detected by qPCR.**

like histamine and leukotrienes and alleviating nasal inflammation and symptoms. A randomized controlled trial conducted by Bavel et al. demonstrated that Beclomethasone is superior to placebo for treating AR and exhibits a safety profile comparable to placebo [58]. In this study, ROC curves were constructed using both training and validation sets to evaluate the diagnostic performance of the model. The results indicated that the area under the curve (AUC) exceeded 0.73, indicating that the immune-related key gene model developed in this study possesses satisfactory diagnostic efficiency. Subsequently, an AR mouse model was successfully established and validated. The expression levels of five genes in the AR model mice were further validated using qPCR experiments, thereby confirming their diagnostic potential.

In summary, this study developed a five-gene model for predicting AR, exhibiting promising prediction accuracy. This model is expected to serve as a foundation for the developing of diagnostic and therapeutic strategies for AR in clinical practice.

## Supporting information

**S1 File. Raw expression profile dataset.**
(RAR)

**S2 File. Batch effect-corrected expression profile dataset.**
(XLSX)

**S3 File. Immune cell proportion dataset with group comparisons.**
(XLSX)

**S4 File. Sample ESTIMATE score dataset (AR vs CTRL).**
(XLSX)

**S5 File. WGCNA co-expression module gene assignment dataset.**
(XLSX)

**S6 File. Significant differentially expressed genes dataset.**
(XLSX)

**S7 File. DEG-WGCNA overlapping genes dataset.**
(XLSX)

**S8 File. Functional enrichment dataset for overlapping DEGs.**
(XLSX)

**S9 File. STRING PPI network edges dataset.**
(XLSX)

**S10 File. PPI network topological analysis scores dataset (MCC/MNC/DEGREE/EPC).**
(RAR)

**S11 File. LASSO diagnostic model coefficients dataset (5-Gene Signature).**
(XLSX)

**S12 File. Training-validation cohort expression profile dataset.**
(XLSX)

**S13 File. Risk-stratified KEGG pathway enrichment dataset.**
(XLSX)

**S14 File. Immune cell-ESTIMATE score correlation matrix dataset.**
(XLSX)

**S15 File. Pharmacochemical compound-gene interaction dataset.**
(XLSX)

## Acknowledgments

We extend our gratitude to the participants of Professor Xuexia Liu's team for their valuable feedback and insightful discussions.

## Author contributions

**Conceptualization:** Maomeng Wang.

**Data curation:** Hua Zhang.

**Formal analysis:** Shuang Wang, XinHua Lin.

**Funding acquisition:** Hua Zhang.

**Investigation:** Shuang Wang, XinHua Lin.

**Methodology:** Maomeng Wang, Shuang Wang.

**Project administration:** Xuexia Liu, Hua Zhang.

**Resources:** XiaoJing Lv.

**Software:** Maomeng Wang.

**Supervision:** Xuexia Liu, Hua Zhang.

**Validation:** Shuang Wang, XinHua Lin, XiaoJing Lv.

**Visualization:** Hua Zhang.

**Writing – original draft:** Maomeng Wang, Shuang Wang.

**Writing – review & editing:** Maomeng Wang, Xuexia Liu.

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
