## [Decision Letter · Decision Letter 0]

19 May 2025

PONE-D-25-10015Identification of immune-related biomarkers associated with allergic rhinitis and development of a sample diagnostic modelPLOS ONE

Dear Dr. Zhang,

Thank you for submitting your manuscript to PLOS ONE. After careful consideration, we feel that it has merit but does not fully meet PLOS ONE’s publication criteria as it currently stands. Therefore, we invite you to submit a revised version of the manuscript that addresses the points raised during the review process.

We look forward to receiving your revised manuscript.

Kind regards,

Shengqian Sun

Academic Editor

PLOS ONE

Journal Requirements:

3. To comply with PLOS ONE submissions requirements, in your Methods section, please provide additional information regarding the experiments involving animals and ensure you have included details on (1) methods of sacrifice, (2) methods of anesthesia and/or analgesia, and (3) efforts to alleviate suffering.

4. Please note that funding information should not appear in any section or other areas of your manuscript. We will only publish funding information present in the Funding Statement section of the online submission form. Please remove any funding-related text from the manuscript.

“This work was supported by Shandong Province Natural Science Foundation youth project (grant number ZR2023QH460) and the Key research and development program of Shandong (grant number 2022CXPT023).”

6. We note that your Data Availability Statement is currently as follows: “All relevant data are within the manuscript and in Supporting Information files.”

Reviewers' comments:

Reviewer's Responses to Questions

**Comments to the Author**

1. Is the manuscript technically sound, and do the data support the conclusions?

Reviewer #1: Yes

Reviewer #2: Yes

2. Has the statistical analysis been performed appropriately and rigorously? 

Reviewer #1: Yes

Reviewer #2: Yes

3. Have the authors made all data underlying the findings in their manuscript fully available?

Reviewer #1: Yes

Reviewer #2: Yes

4. Is the manuscript presented in an intelligible fashion and written in standard English?

Reviewer #1: Yes

Reviewer #2: No

5. Review Comments to the Author

Reviewer #1: This manuscript presents a comprehensive bioinformatics and experimental approach to identify immune-related biomarkers for allergic rhinitis (AR) and to develop a gene-based diagnostic model. The authors successfully integrate transcriptomic data, network analysis, machine learning, and animal model validation. The study is well-structured, methodologically sound, and contributes valuable insights into AR diagnostics and potential therapeutic targets. In my opinion, the manuscript present a well-executed study offering a novel and clinically relevant diagnostic model for AR based on immune-related biomarkers. This manuscript is suitable for publication in is current form.

Reviewer #2: Line 33 and 36. sva and limma need full names or nomenclature. What software have been used for the package? Please be specific in the abstract.

Line 39. SVM and ROC are famous but still need full name at the first time showing up.

Line 23. Although it is the trend fashion using 'We' and it seems for emphasizing the major 'achievement' of a study in generally abstract and conclusion sections, please apply passive transformation for most part of scientific manuscript.

Line 169. Any ethical review protocol has been applied for the mice experiment?

Line 229. 'Error! Reference source not found.' seems something wrong.

Line 254. The author should provide more infomation about Figure 5 volcano plot, what information can be extracted from the volcano plot?

Line 367. substantiates may be too strong for science, probably using indicates or suggests would be better.

Line 380. 'In line with our study,' please indicate which figures or tables are 'in line with' and how?

6. PLOS authors have the option to publish the peer review history of their article (what does this mean? ). If published, this will include your full peer review and any attached files.

**Do you want your identity to be public for this peer review?** For information about this choice, including consent withdrawal, please see our Privacy Policy .

Reviewer #1: **Yes: ** Dr Daniel Elbirt

Reviewer #2: No

---

## [Author Response · Author response to Decision Letter 1]

28 Jun 2025

Dear Editor and Reviewers,

We sincerely appreciate the time and effort you have dedicated to reviewing our manuscript entitled “Identification of immune-related biomarkers associated with allergic rhinitis and development of a sample diagnostic model” (ID: PONE-D-25-10015). Your insightful comments and constructive suggestions have been invaluable in improving the quality and clarity of our work. We have carefully addressed the points raised, and the revisions are highlighted in yellow in the revised manuscript. Below, we provide a detailed response to each reviewer's comments.

Responses to reviews (original comments by reviews are in blue color)

Reviewer # 1:

Comment: This manuscript presents a comprehensive bioinformatics and experimental approach to identify immune-related biomarkers for allergic rhinitis (AR) and to develop a gene-based diagnostic model. The authors successfully integrate transcriptomic data, network analysis, machine learning, and animal model validation. The study is well-structured, methodologically sound, and contributes valuable insights into AR diagnostics and potential therapeutic targets. In my opinion, the manuscript present a well-executed study offering a novel and clinically relevant diagnostic model for AR based on immune-related biomarkers. This manuscript is suitable for publication in is current form.

Reply: We sincerely express our gratitude for your acknowledgment and endorsement of our work.

Reviewer # 2:

1. Comment: Line 33 and 36. sva and limma need full names or nomenclature. What software have been used for the package? Please be specific in the abstract.

Reply: We sincerely appreciate your thoughtful review and valuable feedback on our manuscript. In response we have now consistently supplemented the full nomenclature of the bioinformatics tools “sva” (Surrogate Variable Analysis) and “limma” (Linear Models for Microarray Data) in the abstract. Regarding software implementation, we have explicitly specified the utilization of R packages along with their corresponding versions to ensure complete consistency between the Abstract and Methods sections. (The requested revisions have been implemented in Lines 40-46 of the file labeled 'Revised Manuscript with Track Changes', with all modifications highlighted in yellow for clarity).

2. Comment: Line 39. SVM and ROC are famous but still need full name at the first time showing up.

Reply: Thank you very much for your constructive suggestions. In response, we have thoroughly addressed the requested revisions as follows: In Lines 49-51 of the file labeled 'Revised Manuscript with Track Changes' (highlighted in yellow), we introduced the full terms "Support Vector Machine (SVM)" and "Receiver Operating Characteristic (ROC)" upon their first appearance. To ensure terminological consistency, we have systematically standardized all technical acronyms throughout the manuscript following the "Full Name (Acronym)" format upon initial use, thereby maintaining uniform compliance across all sections.

3. Comment: Line 23. Although it is the trend fashion using 'We' and it seems for emphasizing the major 'achievement' of a study in generally abstract and conclusion sections, please apply passive transformation for most part of scientific manuscript.

Reply: We sincerely thank the reviewer for their valuable guidance on scientific writing conventions. In accordance with this recommendation, we have systematically transformed active constructions into passive voice throughout the manuscript, paying particular attention to the Abstract and Conclusion sections. These revisions ensure complete adherence to the journal's prescribed style for objective academic discourse while maintaining methodological precision.

4. Comment: Line 169. Any ethical review protocol has been applied for the mice experiment?

Reply: Thanks very much for your valuable questions. We confirm full adherence to international standards for the animal research ethics. The study was conducted under ethical approval No, 2025-083 granted by the Animal Experiment Ethics Committee of Yuhuangding Hospital, with all procedures strictly performed in accordance with the approved guidelines. We have provided additional details regarding the animal experiments, including methods of euthanasia and anesthesia, as well as measures implemented to minimize suffering. (The requested revisions have been implemented in Lines 171-173 and Lines 185-190 of the file labeled 'Revised Manuscript with Track Changes', with all modifications highlighted in yellow for clarity).

5. Comment: Line 229. 'Error! Reference source not found.' seems something wrong.

Reply: We sincerely appreciate your suggestion and apologize for the oversight. We confirm that this issue stemmed from a technical artifact caused by figure-format incompatibility during the initial submission process. The revised manuscript now accurately displays all references, as evidenced at Line 236 of the file labeled 'Revised Manuscript with Track Changes' (highlighted in yellow) and consistently throughout the document, with all formatting errors fully resolved.

6. Comment: Line 254. The author should provide more information about Figure 5 volcano plot, what information can be extracted from the volcano plot?

Reply: We sincerely appreciate your detailed comments and suggestions. In Lines 258-264 of the file labeled 'Revised Manuscript with Track Changes' (with all modifications highlighted in yellow), we have supplemented essential information regarding the interpretation of the volcano plot.

7. Comment: Line 367. substantiates may be too strong for science, probably using indicates or suggests would be better.

Reply: We sincerely appreciate this insightful linguistic correction. In Line 363 of the file labeled 'Revised Manuscript with Track Changes' (highlighted in yellow), the verb "substantiates" has been replaced with "suggests" to more precisely convey the inferential nature of our findings.

8. Comment: Line 380. 'In line with our study,' please indicate which figures or tables are 'in line with' and how?

Reply: We sincerely thank you for your constructive suggestion. In Lines 374-378 of the file labeled 'Revised Manuscript with Track Changes' (with all modifications highlighted in yellow), we have revised the text to explicitly establish a connection between our data and the Th2-immunity mechanism, while providing figure-based evidence to support this linkage.

Once again, we extend our sincere gratitude to the reviewers and editors for their invaluable contributions. We believe that the manuscript has been significantly enhanced, and we look forward to its potential publication in your esteemed journal.

Sincerely,

Hua Zhang

---

## [Decision Letter · Decision Letter 1]

18 Jul 2025

Identification of immune-related biomarkers associated with allergic rhinitis and development of a sample diagnostic model

PONE-D-25-10015R1

Dear Dr. Zhang,

We’re pleased to inform you that your manuscript has been judged scientifically suitable for publication and will be formally accepted for publication once it meets all outstanding technical requirements.

Kind regards,

Shengqian Sun

Academic Editor

PLOS ONE

Additional Editor Comments (optional):

Regarding your email query about the clarification of Dr. XueXia Liu’s affiliation, this minor correction should be addressed during the production stage in accordance with PLoS ONE policy. Please notify the production team once your manuscript reaches that phase.

Reviewers' comments:

Reviewer's Responses to Questions

**Comments to the Author**

1. If the authors have adequately addressed your comments raised in a previous round of review and you feel that this manuscript is now acceptable for publication, you may indicate that here to bypass the “Comments to the Author” section, enter your conflict of interest statement in the “Confidential to Editor” section, and submit your "Accept" recommendation.

Reviewer #2: All comments have been addressed

2. Is the manuscript technically sound, and do the data support the conclusions?

Reviewer #2: Yes

3. Has the statistical analysis been performed appropriately and rigorously? 

Reviewer #2: Yes

4. Have the authors made all data underlying the findings in their manuscript fully available?

Reviewer #2: Yes

5. Is the manuscript presented in an intelligible fashion and written in standard English?

Reviewer #2: Yes

6. Review Comments to the Author

Reviewer #2: The authors have significantly improved the manuscript, and all key issues have been addressed, so I have no further questions.

Cheers,

7. PLOS authors have the option to publish the peer review history of their article (what does this mean? ). If published, this will include your full peer review and any attached files.

**Do you want your identity to be public for this peer review?** For information about this choice, including consent withdrawal, please see our Privacy Policy .

Reviewer #2: No

---

## [Editor Report · Acceptance letter]

PONE-D-25-10015R1

PLOS ONE

Dear Dr. Zhang,

I'm pleased to inform you that your manuscript has been deemed suitable for publication in PLOS ONE. Congratulations! Your manuscript is now being handed over to our production team.

Kind regards,

on behalf of

Dr. Shengqian Sun

Academic Editor

PLOS ONE